# Substantial lifetime enhancement for Si-based photoanodes enabled by amorphous TiO₂ coating with improved stoichiometry

Yutao Dong[1], Mehrdad Abbasi[2], Jun Meng[1], Lazarus German[1], Corey Carlos[1], Jun Li[1], Ziyi Zhang [1], Dane Morgan [1], Jinwoo Hwang[2] & Xudong Wang [1]✉

Amorphous titanium dioxide (TiO₂) film coating by atomic layer deposition (ALD) is a promising strategy to extend the photoelectrode lifetime to meet the industrial standard for solar fuel generation. To realize this promise, the essential structure-property relationship that dictates the protection lifetime needs to be uncovered. In this work, we reveal that in addition to the imbedded crystalline phase, the presence of residual chlorine (Cl) ligands is detrimental to the silicon (Si) photoanode lifetime. We further demonstrate that post-ALD in-situ water treatment can effectively decouple the ALD reaction completeness from crystallization. The as-processed TiO₂ film has a much lower residual Cl concentration and thus an improved film stoichiometry, while its uniform amorphous phase is well preserved. As a result, the protected Si photoanode exhibits a substantially improved lifetime to ~600 h at a photocurrent density of more than 30 mA/cm². This study demonstrates a significant advancement toward sustainable hydrogen generation.

Photoelectrochemical (PEC) water splitting is considered a promising sustainable energy path to hydrogen fuel generation[1]. Silicon (Si) is a commonly used photoanode material that is expected to deliver the highest energy conversion efficiency owing to its appropriate band structure and excellent charge mobility[2,3]. However, Si also exhibits a very short lifetime of only a few hours, as it would experience rapid corrosion or dissolution in alkaline electrolytes[4]. To improve its lifetime, a prevailing strategy is to protect the reactive photoanode surfaces with a conformal and pinhole-free inert oxide coating by atomic layer deposition (ALD)[5-7]. While both crystalline and amorphous ALD oxide films have been applied for surface protection[8], amorphous coating are more commonly used owing to its excellent uniformity and conformality. The highest reported lifetime of Si-based photoanode was achieved by amorphous TiO₂ coating at the level of 500 h[2,6,8-12], which, however, is still far from the minimal industrial requirement (30,000 h)[13]. Further improving the lifetime appeared to be stunted. A wide variety of performances have been reported from ALD coatings from different growth systems or slightly changed deposition conditions[9,11,14,15], limiting the reproducibility and scaling up of this promising strategy in energy applications. Other ALD oxide films, such as Al₂O₃, HfO₂, and ZrO₂, have also been explored as a protective coating. Their protection performances were generally lower than TiO₂, possibly due to their lower conductivity and structural uniformity[10,16,17].

Despite the broad use of ALD for surface protection, the essential structure-property relationship that dictates the protection lifetime control is still largely missing. Typically, the amorphous ALD films are considered homogeneous. Their protection performances are primarily tuned by thickness and deposition temperature. Among the limited understandings of the protection lifetime control, a few reports correlated the protection failure with extrinsic impurities or nanoparticles[14,15,18-20]. They revealed that failure typically starts from pinhole formation, which is consistent across different electrode systems. However, current studies still cannot fully answer the fundamental question, i.e., what is the correlation between ALD process, structure, and chemistry of the amorphous films, and their chemical and physical properties, which determines the protection performance? This knowledge gap places the main hindrance to further improving the PEC electrode lifetime to meet the industrial standard. Our previous study

[1]Department of Materials Science and Engineering, University of Wisconsin-Madison, Madison, WI 53706, USA. [2]Department of Materials Science and Engineering, The Ohio State University, Columbus, OH 43210, USA. ✉e-mail: xudong.wang@wisc.edu

revealed that structural inhomogeneity in the amorphous film (e.g., imbedded intermediate phases) could induce a highly localized current through the amorphous film and facilitate pinhole formation[9]. While this study suggested a low-temperature crystalline-free amorphous film is preferable for achieving a long lifetime, the unreacted precursor ligands and byproducts were another inevitable issues associated with the low-temperature ALD processes. These impurities could substantially change the film properties, such as electronic and ionic transport properties, mechanical stability, and chemical reactivity[21–24], which may open another pathway to breaking the current protection lifetime limit. In this paper, we report a fundamental discovery of the detrimental role of the residual Cl ligand in amorphous $TiO_2$ films to the PEC protection lifetime. We further developed a post-ALD in situ water treatment procedure. It was able to partially remove the Cl residues and improve the film stoichiometry without introducing any additional crystallization. The processed amorphous $TiO_2$ film showed drastically improved chemical stability, and thereby substantially elongated the lifetime of Si photoanodes to ~600 h in an alkaline electrolyte. The result will bring direct impacts to the next-generation solar fuel production devices with enhanced lifetime and stability.

## Results

### Failure of Amorphous ALD TiO₂ coating in PEC protection

The typical configuration of a Si-based photoanode is illustrated in Fig. 1a. The surface of the n-type Si photoabsorber was coated by an ultrathin film of amorphous $TiO_2$ via ALD, followed by a layer of Ni acting as the oxygen evolution reaction (OER) catalyst. The conformal amorphous $TiO_2$ coating prohibits OH⁻ group diffusion thus protects the Si surface from chemical corrosion. The ALD $TiO_2$ layer can also passivate Si surface defect states to suppress charge recombination at the interface[25]. Meanwhile, the ultrathin film still permits adequate hole transport through defect band conduction, allowing for unimpaired PEC efficiency[11,26]. Owing to the excellent conformality of ALD, the as-deposited $TiO_2$ showed an extremely clear and smooth surface without any observable particle features (Fig. 1b). Atomic force microscopy (AFM) topography scan revealed that the average surface roughness of $TiO_2$ was only 0.33 nm, confirming the outstanding smoothness and uniformity (Fig. 1c). The structure of as-deposited $TiO_2$ films were examined by scanning transmission electron microscopy (STEM). Figure 1d shows the cross-section of a 15-nm-thick $TiO_2$ film on Si, which was free of any particles. The nano-diffraction pattern of the $TiO_2$ film indicated a completely amorphous phase without any crystalline phases (Inset of Fig. 1d)[9]. X-ray photoelectron spectroscopy (XPS) survey spectrum showed strong Ti and O peaks confirming the film's chemical composition (Fig. 1e). In addition, an appreciable Cl signal was also detected. The existence of Cl elements could be attributed to the unreacted Cl ligand from the $TiCl_4$ precursor at a relatively low deposition temperature (100 °C)[27].

The high homogeneity of the amorphous $TiO_2$ film without any observable crystalline phases is a desired coating feature that is

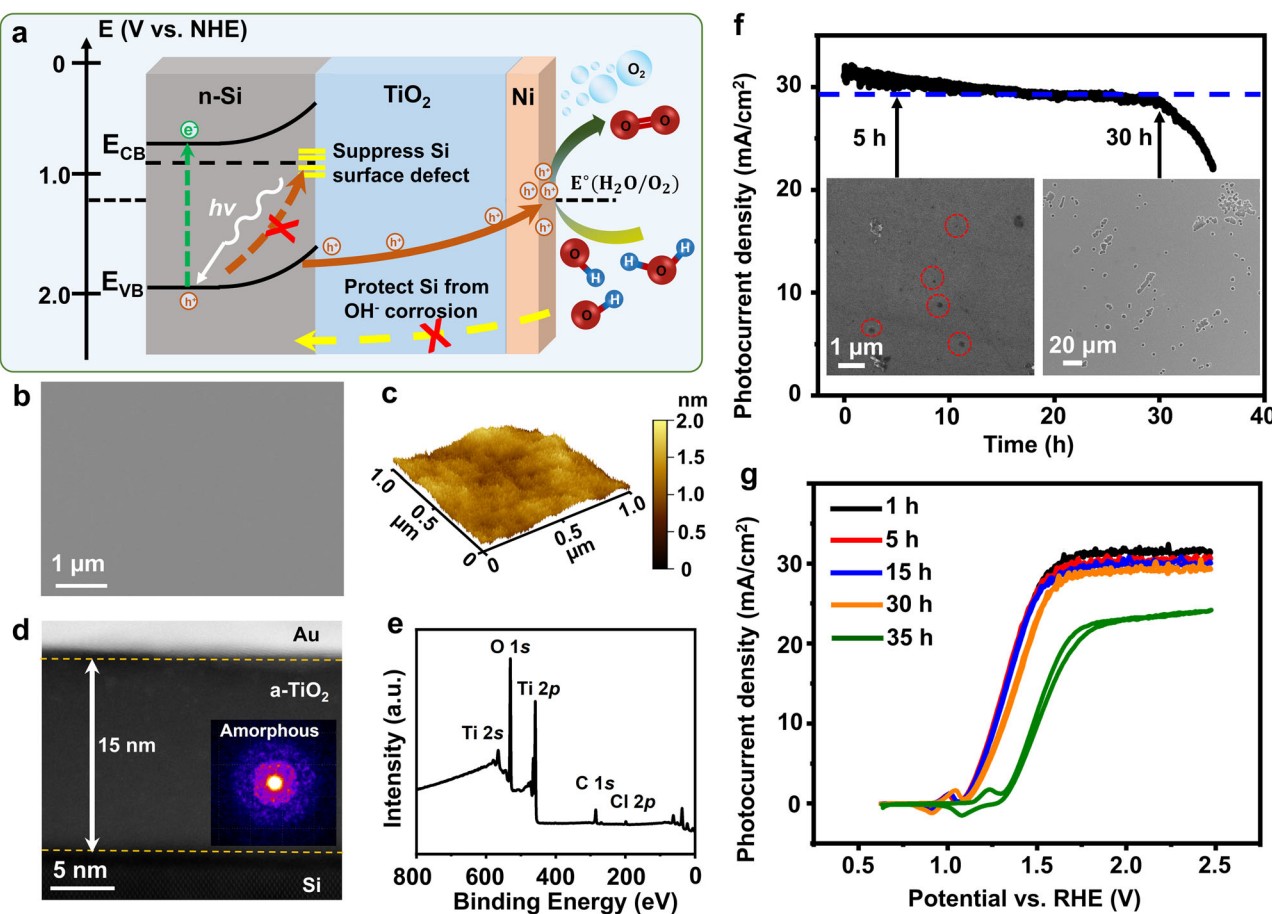

**Fig. 1 | Si photoanodes protected by amorphous TiO₂ thin film coating.**
**a** Schematics of a Si/TiO₂/Ni photoanode for PEC water oxidation. The relevant electronic energy levels in different materials are schematically shown. NHE normal hydrogen electrode, CB conduction band, VB valence band. **b** SEM image of as-deposited amorphous TiO₂ thin film on n-Si substrate. **c** AFM topography of the TiO₂ surface. **d** Cross-sectional STEM image of Si/TiO₂. Inset is a nano-diffraction pattern taken from the TiO₂ region. **e** XPS survey spectrum of the TiO₂ film on Si wafer. **f** Chronoamperometry test of Si/TiO₂/Ni photoanode measured in 1.0 M KOH aqueous solution under 1 sun illumination at an external bias of 1.8 V vs. RHE. The blue line marks 90% of the original $J_{ph}$ value. Insets show Si/TiO₂/Ni surfaces after 5 h (left) and 30 h (right) chronoamperometry test. **g** $J_{ph}$-$V$ curves of Si/TiO₂/Ni photoanode obtained at a series of time points from the chronoamperometry test.

expected to bring high stability and a long lifetime to the Si-based PEC system in alkaline electrolyte[8,9]. To reveal the protection performance of this homogeneous amorphous $TiO_2$ thin film, the complete n-Si/$TiO_2$/Ni photoanode was placed in a 1 M KOH aqueous solution under 1 sun illumination for PEC water oxidation tests. It should be noted that the sputtered Ni catalyst film remained flat and did not introduce any damage to the $TiO_2$ surface (Figs. S1, S2). They were electrochemically converted to $Ni(OH)_2$ sheet-like structure during the PEC test (Fig. S3). The PEC performance was investigated in a three-electrode system. Chronoamperometry test at a bias of 1.8 V versus reversible hydrogen electrode (RHE) revealed a quick photocurrent density ($J_{ph}$) decay (Fig. 1f), where the original value dropped by 10% within just 30 h. At the point of failure (defined as the time point when $J_{ph}$ reached <90% of its original value), a large number of interconnected pores appeared on the electrode surface (right inset of Fig. 1f and Fig. S4a), suggesting low stability of this $TiO_2$ coating. These pores were evolved from small pinholes as early as a few hours of operation, while $J_{ph}$ was still >95% of its original value (Left inset of Fig. 1f and Fig. S4b). The large pores would isolate the Si photo-absorber from the Ni catalyst layer, and facilitate the formation of insulating $SiO_x$ that limits the hole transport from Si to Ni catalyst. As a result, the Si/$TiO_2$/Ni photoanode PEC performance was impaired. The $J_{ph}$ vs. potential ($V$) curves were recorded at a few time points through the chronoamperometry test (Fig. 1g). In the first hour, the water oxidation onset potential was 1.08 V versus RHE and $J_{ph}$ reached a saturated value of 31.5 mA/cm$^2$ at 1.8 V versus RHE. The saturated $J_{ph}$ and onset potential were on par with reported benchmark n-Si-based photoanodes[11,16,28–30], indicating the high quality of our Si/$TiO_2$/Ni system. This onset potential was kept steady for the first 15 h, but quickly shifted positively to 1.11 and 1.24 V versus RHE at the 30-h and 35-h operation time points, respectively (Fig. 1g). The higher onset potential implied the increase of charge transfer resistance in the PEC system, which is typically induced by the formation of insulating $SiO_2$ layer due to Si corrosion[31,32]. Accordingly, saturated $J_{ph}$ dropped continuously to 30.2, 29.8, 28.8, and 21.9 mA/cm$^2$ at the PEC operation time of 5, 15, 30, and 35 h, respectively, consistent with the decay trend in the stability test. During PEC operation, because no extra redox peaks other than the $Ni(OH)_2$/NiOOH couple were observed from the $J_{ph}$-$V$ curves, it can be hypothesized that the corrosion of the $TiO_2$ layer was a result of chemical dissolution without valence change.

To test this hypothesis, the $TiO_2$-coated Si wafer was immersed into a 1 M KOH aqueous solution to evaluate the chemical stability of the amorphous $TiO_2$ film. Without an external bias, the $TiO_2$-coated Si wafer still exhibited obvious corrosion but with square-like pores, a typical morphology of chemically corroded Si wafer in an alkaline solution (Fig. S5)[18,33]. This type of corrosion was also started from pinholes (Fig. S5a–c) as a result of $TiO_2$ dissolution confirmed by XPS analyses (Fig. S6a–c), suggesting the chemical reactivity of as-deposited $TiO_2$ would primarily be responsible for the failure. Considering the presence of an appreciable amount of precursor ligand residues (Cl from $TiCl_4$ precursor, Fig. 1e) in the film, the unreacted Cl ligands acted as the terminating point in the Ti-O-Ti network, and thus could introduce a more permeable amorphous lattice allowing fast reaction between $OH^-$ and Ti-O[34]. In addition, DFT calculation revealed that residual Cl ligands could also induce in-gap defect states at the middle of the band gap and shifted the Fermi level from 2.71 eV (Fig. 2a) to 4.14 eV (Fig. 2b) closer to the conduction band, which can enhance the hole conductivity of the amorphous $TiO_2$ film[12,26]. This localized conductivity increase is able to induce local $OH^-$ accumulation at the electrode surface, which facilitates the reaction between $OH^-$ and $TiO_2$[9]. Together, in the PEC system, Cl residues could enable a faster dissolution of $TiO_2$ and a rapid diffusion of $OH^-$ to reach the vulnerable Si surface (Fig. 2c)[35–37].

## Water-treated amorphous $TiO_2$ with improved stoichiometry

Due to the destructive residual Cl ligands, therefore, it is intuitive to hypothesize that a longer protection lifetime may be achieved by reducing the residual Cl ligands in amorphous $TiO_2$ film. Meanwhile, no crystallization should be induced, as crystalline phases can introduce structural inhomogeneity and jeopardize the film's lifetime. To achieve this goal, a post-ALD water treatment procedure is developed to reduce the residual Cl ligands and maintain amorphous film homogeneity simultaneously. As schematically illustrated in Fig. 3a, after the regular $TiO_2$ ALD cycles are completed, the sample was kept in the growth chamber subjecting to additional 2400 water pulses under the same temperature and vacuum. During this treatment, water molecules could diffuse into the amorphous $TiO_2$ matrix and react with the dangling Cl ligands, which raises the film's stoichiometry and improves the Ti-O-Ti network continuity. This post-ALD water treatment could well preserve the surface flatness and conformality of the as-deposited $TiO_2$ film. No additional nanoparticles were observed from the film surface (Fig. 3b). AFM topography scan revealed that the $TiO_2$ surface kept the same extremely low roughness of ~0.3 nm (Fig. 3c). XPS Cl 2$p$ spectra showed an apparent intensity drop of both Cl 2$p_{3/2}$ (198.16 eV) and Cl 2$p_{1/2}$ (199.82 eV) peaks (corresponding to the Cl-Ti bonding[27,38]) after water treatment (Fig. 3d). The fine Ti 2$p$ spectra of both pristine and water-treated films showed an almost identical shape with the dominating Ti$^{4+}$ chemical state located at 459 eV (Fig. 3e), implying the water treatment did not change the chemical state of Ti$^{4+}$ cation in the network. By integrating the Cl, O, and Ti peaks areas as a half-quantitative analysis, the Cl:Ti ratio was found reduced by 27% (from 0.062 to 0.045), confirming a substantial removal of residual Cl ligands from the amorphous $TiO_2$ film.

Four-dimensional scanning electron microscopy (4D-STEM) based characterizations were further used to confirm the elemental and structural change from water treatment. The presence of two blurry rings in the averaged nano diffraction patterns revealed that no additional crystallization was induced by the water treatment (Fig. 4b) as compared to the pristine $TiO_2$ film (Fig. 4a). Diffraction also suggested the existence of the medium-range ordering (MRO) in both $TiO_2$ amorphous films. MRO refers to the nanoscale volumes with relatively high structural ordering within amorphous materials[39]. Here, the variance ($V$) of the nano diffraction intensity as a function of reciprocal lattice vector $k$, $V(k)$ was used to measure the degree of MRO (Fig. 4c). The magnitude of the peak is related to the degree of structural fluctuation created by the distribution of MRO domains, and the peak position is related to the type of MRO. The broad peak at ~3.0 nm$^{-1}$ indicates the MRO was substantially disordered, confirming the amorphous phase. After water treatment, the broad peak position was retained without shifting, demonstrating disordered MRO structure was well maintained. The slightly increased peak intensity may be related to Cl ligands removal. The change of Cl distribution was directly visualized using long-time energy dispersive spectroscopy (EDS) Cl mapping acquired from the cross-section of both $TiO_2$ film samples. Compared to the uniform distribution of the Cl signal in the pristine $TiO_2$ film (Fig. 4d), the Cl signal in the water-treated $TiO_2$ film was clearly decreased with a concentration gradient, where the Cl signal was nearly undetectable from the top ~5 nm region. EDS spectrums were then collected from the top and bottom regions (marked by red dashed boxes) to quantify the location-dependent chemical composition change (all peak intensities were normalized by the Ti K peaks for comparison). Comparing the water-treated $TiO_2$ film (Fig. 4f) to the pristine $TiO_2$ film (Fig. 4e), Cl:Ti ratio in the bottom region was reduced from 3.86 to 2.41% and in the top region was reduced from 2.46 to 1.81%. These characterizations further confirmed that post-ALD water treatment was able to partially remove residual Cl ligands without introducing additional crystallization to the amorphous ALD $TiO_2$ films.

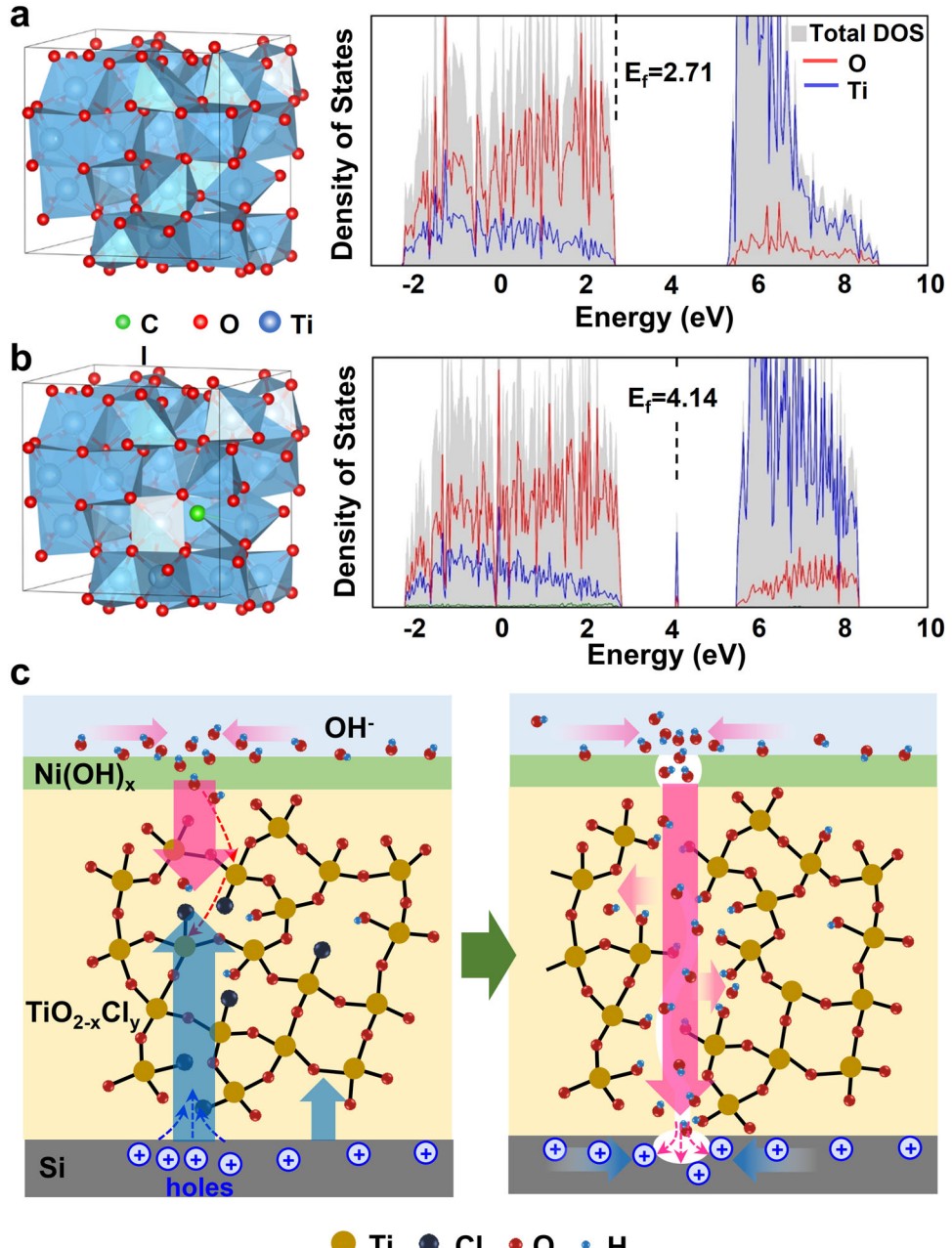

**Fig. 2 | Cl-related TiO₂ failure mechanism. a, b** Density of states of the melt-quenched amorphous TiO₂ and TiO$_{1.965}$Cl$_{0.035}$ models calculated by DFT. The left insets are the polyhedral visualization of the structure of amorphous TiO₂ and TiO$_{1.965}$Cl$_{0.035}$. **c** Schematics of a possible mechanism of pinhole formation in Cl-containing amorphous TiO₂ films during PEC water oxidation.

## PEC stability by water-treated amorphous TiO₂ coating

The chemical stability of water-treated TiO₂ films was then evaluated by immersing the treated Si/TiO₂ sample in a 1 M KOH aqueous solution without applying any external bias. Compared to pristine TiO₂-coated Si, the density of square-like corrosion spots were substantially reduced from ~300 to ~30 mm$^{-2}$, with smaller spot sizes (Fig. S5). XPS elemental analysis revealed nearly identical Cl 2$p$, Ti 2$p$, and O 1$s$ peak intensities after immersion (Fig. S6d–f), indicating that the TiO₂ film after water treatment could be largely preserved when facing an alkaline solution. Electrochemical impedance spectroscopy (EIS) of Si/TiO₂ electrodes was applied to study the possible TiO₂ film conductivity change induced by Cl removal (Fig. 5a). The Nyquist plots were fitted based on the equivalent circuit model (Inset of Fig. 5a). The Si/TiO₂ electrodes with and without water treatment showed similar charge transfer resistance at the high-frequency region, suggesting partially removing Cl did not impair the charge transport property of the TiO₂ film.

Above characterizations confirmed that post-ALD water treatment was able to largely improve the chemical stability and preserve good electrical conductivity. To further evaluate its influences on the PEC performance, the same amount of Ni catalysts was deposited on Si/TiO₂ for water oxidation reaction under the same conditions. Figure 5b compares the $J_{ph}$-$V$ curves of Si/TiO₂/Ni photoanodes with and without water treatment. Both curves exhibited similar onset potential and saturated $J_{ph}$. The slightly increased slope of the $J_{ph}$-$V$ curve from the water-treated TiO₂ sample suggests there might be less amount of charge recombination due to the improved charge transport property[40]. With the improved PEC performance, the

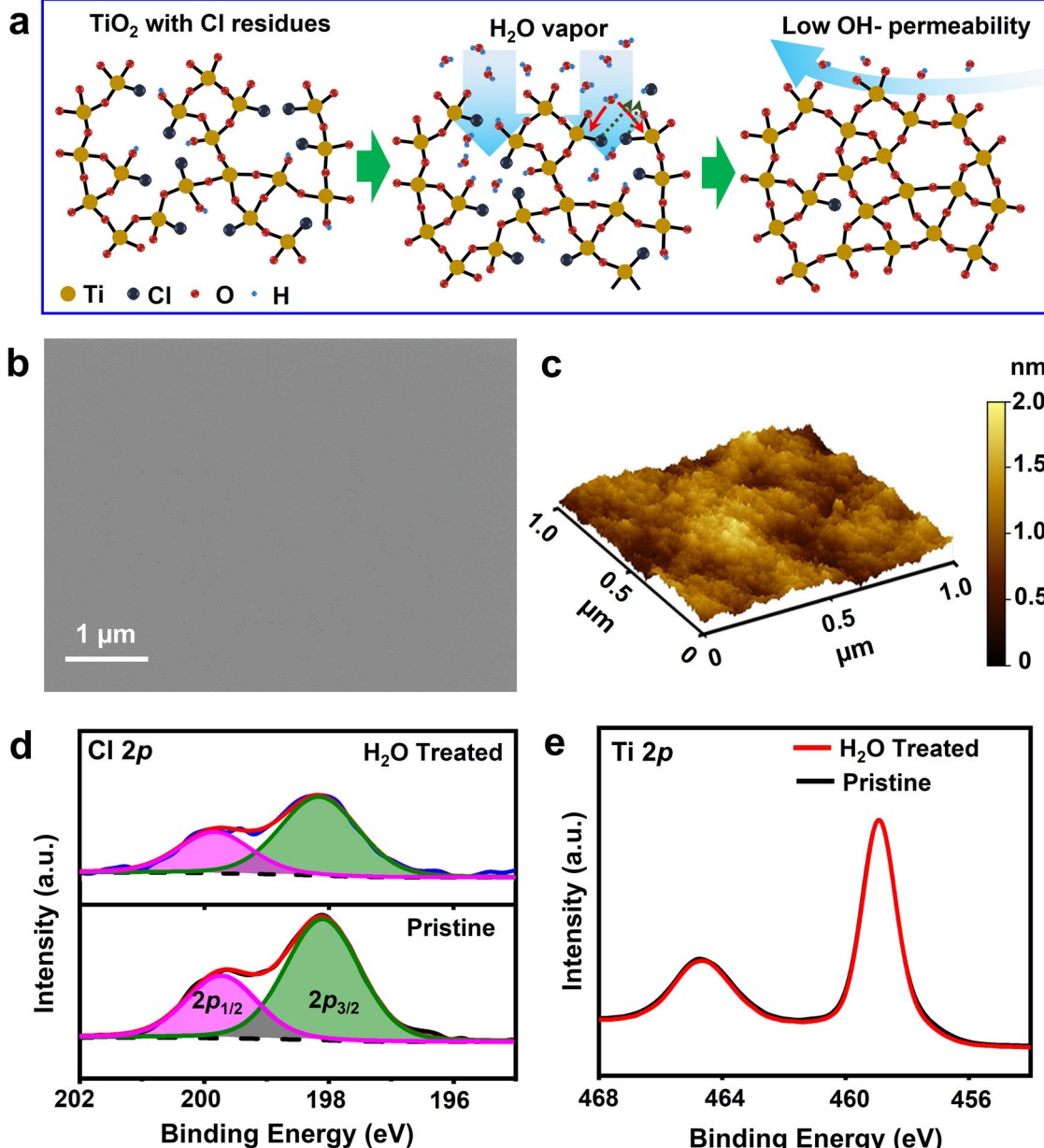

**Fig. 3 | Water treatment of amorphous TiO₂ films. a** Schematics of water treatment showing the unreacted Cl ligands in amorphous TiO₂ (Left) being removed by reacting with $H_2O$ molecules diffused through the film surface (center) and forming a more continuous, interconnected Ti-O-Ti network. **b** SEM image of the amorphous TiO₂ film after being treated under pulsed water vapor exposure. **c** AFM topography of the water-treated TiO₂ surface with nearly the same surface roughness as the pristine films. **d, e** XPS core spectra of Cl 2*p* (**d** red and green area are integrated Cl 2$p_{1/2}$ and 2$p_{3/2}$ peak. Black dash lines are baselines for peak area integration.) and Ti 2*p* (**e**) comparison between pristine and water-treated TiO₂ thin films.

chronoamperometry test was conducted at an external bias of 1.8 V versus RHE. A stable $J_{ph}$ at ~30 mA/cm² was recorded for up to ~600 h (Fig. 5c), about one order of magnitude longer than the pristine TiO₂-protected Si photoanodes. $J_{ph}$-$V$ curves were also recorded at a series of reaction time points to understand the PEC property change during this long operation period (Fig. 5d). The $J_{ph}$-$V$ curve maintained an identical shape for the first >100 h, where the electrode surface was nearly intact showing extremely high stability (Fig. 5e-i). The slope started to show a subtle decrease till ~250 h. By that time, only a few small pinholes were evolved on the electrode surface (Fig. 5e-ii). As the reaction time extended, the Ni(OH)₂/Ni(OOH) redox peaks and the saturated potential gradually shifted anodically, and the $J_{ph}$-$V$ curve slope slightly decreased. Both can be attributed to the increase of charge transfer resistance due to the formation of

SiOₓ on the Si surface. The saturated $J_{ph}$ was almost maintained at the same ~30 mA/cm² throughout the entire testing period, evidencing the long-term high water oxidation performance. Owing to the much higher chemical stability of the water-treated TiO₂ films, the evolution from several pinholes to large interconnected pores was substantially suppressed. Therefore, the Si photoabsorber and the TiO₂/Ni catalyst layer could still maintain a tight connection and allow nearly impaired charge flow before the formation of large pores. When the photoanode approached its failure point at 600 h, the $J_{ph}$-$V$ curve exhibited a drastic anodic shift and the saturated photocurrent density decreased apparently. At this point, large and interconnected pores would be observed on the electrode surface (Fig. 5e-iii), indicating the water-treated TiO₂ film may still share the same failure mechanism as the pristine TiO₂ films.

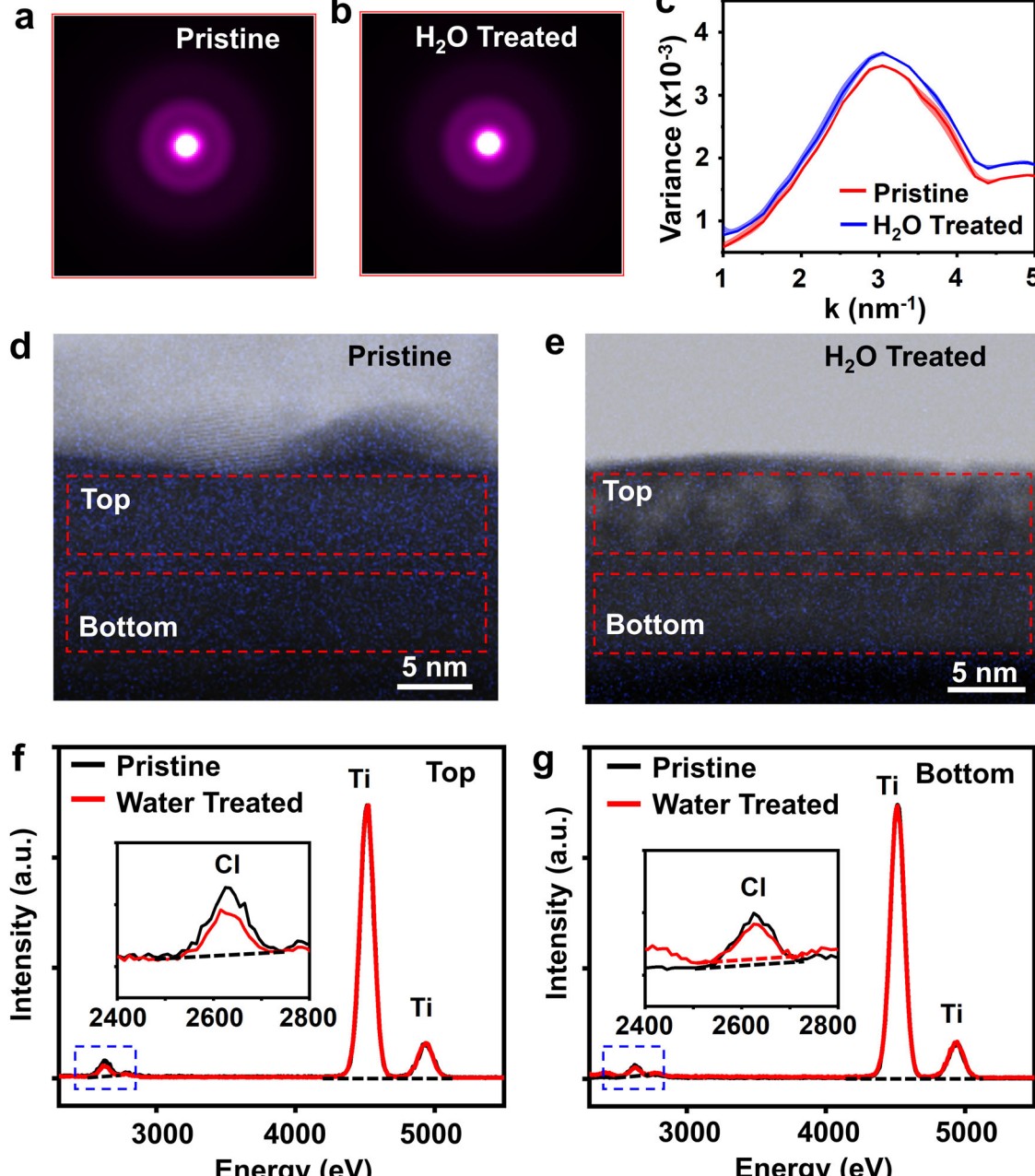

**Fig. 4 | 4D-STEM characterization and comparison of pristine and water-treated TiO₂ films. a, b** Averaged 4D-STEM nano-diffractions of pristine (**a**) and water-treated (**b**) TiO₂ film, showing both films had a pure amorphous phase without any crystalline impurities. **c** Comparison of the experimental $V(k)$ between pristine (red) and water-treated (blue) TiO₂ films. Shade areas represent the uncertainty range of $V(k)$. **d, e** Cross-sectional EDS Cl mapping of pristine and water-treated TiO₂ films supported on Si wafers. Red dash boxes defined the top and bottom regions in the cross sections, where EDS signals were collected. **f, g** EDS spectra of pristine and water-treated TiO₂ collected from the top (**f**) and bottom (**g**) regions from the cross sections. (peaks were normalized by the Ti K peaks). Insets are enlarged Cl peaks in the blue dash box to show the intensity change after water treatment. Black dash lines are baselines for peak area integration.

## Discussion

From the above studies, the excellent PEC stability could be attributed to the reduced Cl concentration and improved film stoichiometry without introducing an additional crystalline phase in the amorphous TiO₂ coating, a key structural feature enabled by the post-ALD water treatment. This was because when the ALD growth was completed, the 3D amorphous TiO₂ matrix was formed, and the completely coordinated Ti-O largely restricted their diffusion to form an ordered TiO₂ crystal lattice. It is important to point out that this post-ALD water treatment is fundamentally different from other regular approaches to improve the stoichiometry of ALD films. Reducing the residual ligands

(i.e., raising the ALD reaction completeness to improve the stoichiometry) and suppressing the crystallization are always coupled and anticorrelated in regular ALD processes. For example, extending the length of the ALD water pulse can improve the reaction completeness per cycle. However, the extended reaction time could also facilitate surface diffusion of as-deposited species that were loosely bonded to the surface. This may enable the rearrangement of surface atoms and form local nuclei on the surface[41–44], which could serve as seeds for TiO₂ to grow into nanoparticles. As shown in Fig. S7, after elongating the water pulse time from 0.5 to 6.5 s, Cl 2p core spectrum showed an apparent peak intensity drop, validating the improved ALD reaction

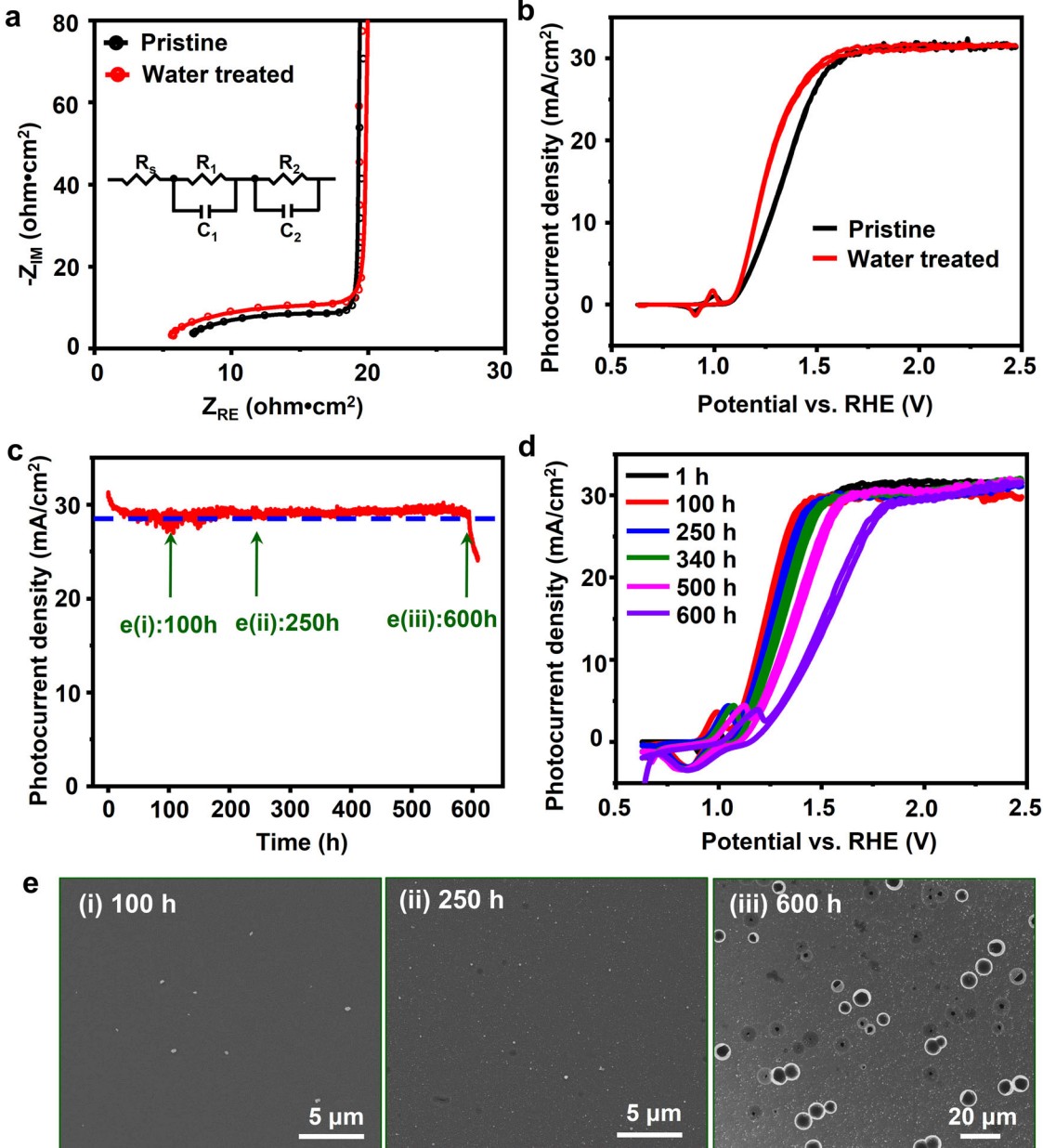

**Fig. 5 | Lifetime evaluation of water-treated TiO₂ film for Si photoanodes protection. a** EIS measurement of photoanodes protected by pristine and water-treated TiO₂ films under 1 sun illumination. Inset is the equivalent circuit for curve fitting. **b** $J_{ph}$-$V$ curves of Si/TiO₂/Ni photoanodes protected by pristine (black) and water-treated (red) amorphous TiO₂ films. **c** Chronoamperometry of the Si/TiO₂/Ni electrode (with water treatment) measured in 1.0 M KOH aqueous solution under 1 sun illumination at an external bias of 1.8 V vs. RHE. The blue line marks 90% of the original $J_{ph}$ value. **d** $J_{ph}$-$V$ curves of Si/TiO₂/Ni photoanode (with water treatment) obtained at a series of time points from the chronoamperometry test. **e** SEM images of the photoelectrode surfaces collected at 100 h (i), 250 h (ii), and 600 h (iii) during the chronoamperometry test, revealing the very slow surface corrosion.

completeness. However, the as-prepared TiO₂ film exhibited a large number of nanoparticles on the surface and the corresponding structure heterogeneity would yield an impaired protection performance (Fig. S8a, b). Similarly, due to the lower energy barrier for nucleating, particularly the intermediate phases of TiO₂ at the active growth surfaces[9], other regular approaches to achieving complete ALD reactions, such as raising the temperature or introducing plasma, are always associated with undesirable crystallization, which also jeopardizes the protection lifetime as earlier research discovered[38,45]. For example, TiO₂ film deposited under 160 °C yielded a large number of crystalline nanoparticles in the amorphous matrix (Fig. S9a). The corresponding TiO₂-protected Si photoanode only showed a ~70 h lifetime for PEC operation (Fig. S9b).

The critical role of water vapor exposure in the post-ALD treatment is further demonstrated by annealing the pristine TiO₂ films in an N₂ atmosphere or under TiCl₄ exposure instead. Annealing TiO₂ films in the N₂ atmosphere showed almost the same peak intensity in XPS Cl 2$p$, Ti 2$p$ core spectrum as compared to pristine TiO₂ film (Fig. S10a, b), suggesting the inert gas atmosphere wouldn't change the chemical composition of the ALD film. Thus, the PEC performance of N₂-annealed TiO₂ film had a similar lifetime performance as pristine TiO₂ (10% decay within 29 h, Fig. S10c). Annealing Si/TiO₂ in the TiCl₄ atmosphere brought extra Cl impurities to the film and exhibited acute chemical instability. A large amount of corrosion spots emerged after 1 day of immersion (Fig. S11a−c). The drastic drop of peak intensity in XPS Cl 2$p$, Ti 2$p$, and O 1$s$ spectra demonstrated the quick dissolution

of $TiO_2$ from the film (Fig. S11d–f). Correspondingly, $TiCl_4$-treated $TiO_2$ film showed a very short PEC protection lifetime with 10% photocurrent decay in only 1 h (Fig. S12). These control experiments also proved the critical role of Cl impurities in determining the chemical stability and protection lifetime of ALD $TiO_2$ films.

Besides $TiCl_4$ precursor, tetrakis(dimethylamido)titanium (TDMAT) is another typical Ti source for ALD of $TiO_2$ film. However, growing $TiO_2$ by TDMAT at the same low temperature (100 °C) would leave a substantial amount of molecular ligand residues. These big residual molecules could reduce $TiO_2$ film density and even result in pores formation[46,47]. Therefore, ALD of $TiO_2$ was conducted under the recommended temperature ($T = 250$ °C) using TDMAT as Ti precursor to understand the precursor relationship with the water treatment strategy and corresponding PEC performance. The as-grown $TiO_2$ film clearly showed an appreciable amount of nanoparticles on the surface, suggesting crystallization already occurred during the growth (Fig. S13a). However, the water treatment did not introduce an obvious change to the surface nanoparticles (Fig. S13b). From XPS spectra, a small N peak was detected from both $TiO_2$ films before and after water treatment, with just an infinitesimal change in its intensity (Fig. S13c). Integrating the N and Ti peak areas revealed a much smaller N:Ti ratio reduction of ~9% compared to that of Cl:Ti ratio (27%) from the $TiCl_4$ precursor. This low removal ratio of TDMAT residues could be attributed to the limited diffusion as a result of the high steric hindrance from the large dimethylamido ligands. Accordingly, the lifetime of $TiO_2$ films made from TDMAT was only slightly extended from ~150 to ~190 h after the water treatment (Fig. S13e), much shorter than the ~600 h offered by water-treated $TiO_2$ films made from $TiCl_4$. This result further evidenced that TDMAT may not be the optimal choice for ALD of protective $TiO_2$ films, as it introduces crystalline nanoparticles under conditions necessary to achieve sufficiently high ALD reactions. This result provided additional support to our hypothesis that crystal-free is essential to achieve a long protection lifetime, and small molecule $TiCl_4$ precursor is the optimal choice to achieve this goal.

In conclusion, we showed that post-ALD water treatment could partially remove the unreacted Cl ligands in ALD amorphous $TiO_2$ films without introducing additional crystallization, thereby largely improving the film's lifetime for Si-photoelectrode protection. In amorphous $TiO_2$ films deposited at low temperatures, residual ligands are inevitable. Their presence could significantly increase the film reactivity in alkaline electrolytes, resulting in pinhole formation and quick dissolution. Removing the residual Cl ligands by completing the reaction with extra water exposure led to the closer-to-ideal stoichiometry of $TiO_2$ film with improved Ti-O-Ti network continuousness. The amorphous film structure was well retained during this extended ALD process, possibly due to the limited mobility of Ti-O polyhedrons inside the amorphous matrix. When applied as a Si photoanode protection layer, this homogeneous amorphous $TiO_2$ film exhibited an ultra-stable protection performance in an alkaline solution, maintaining a very high saturated $J_{ph}$ at 30 mA/$cm^2$ for ~600 h. This discovery provided a promising solution to decouple the crystallization from raising the ALD reaction completeness. The nearly perfect amorphous ALD film with controlled stoichiometry may enable an essential manufacturing capability leading the PEC photoelectrodes to meet the industrial standard.

## Methods

### ALD synthesis of $TiO_2$ thin films

The n-type Si wafers in our experiments were 380-μm-thick, 3-inch-diameter, single-side polished, <100> oriented, and with a resistivity of 1–10 Ω·cm. Prior to ALD, Si wafers were washed with acetone, isopropanol, and deionized (DI) water in an ultrasonic bath for 20 min sequentially, followed by immersing in 5 wt% HF solution to remove the native oxide. $TiO_2$ was deposited in a homemade ALD system following reported procedures[2,9]. In specific, $N_2$ gas with a flow rate of 40 sccm was introduced into the chamber to serve as the carrier gas. The system base pressure was kept at 780 mTorr. The chamber temperature was maintained at 100 °C for depositions. Precursors used for $TiO_2$ deposition were $TiCl_4$ (Sigma-Aldrich, 99.9%) and DI $H_2O$. $TiCl_4$ (Sigma-Aldrich, 99.9%). Both precursor vapors were pulsed into the deposition chamber separately with a pulsing time of 0.5 s each and separated by 60 s $N_2$ purging. Therefore, one deposition cycle involves 0.5 s of $H_2O$ pulse + 60 s of $N_2$ purging + 0.5 s of $TiCl_4$ pulse + 60 s of $N_2$ purging. Through this procedure, ~15 nm $TiO_2$ film was obtained after 200 cycles. For TDMAT-$TiO_2$ film, the film was deposited under a recommended temperature of 250 °C in Fiji G2 ALD with TDMAT precursor (Sigma-Aldrich, 99.99%) with 300 cycles for comparison.

### Post-ALD water treatment

Immediately after the normal ALD procedure being completed, additional 2400 water pulses were introduced to the ALD chamber. Each water pulse was separated by 10 s $N_2$ purging. The other chamber conditions remained the same. After this water treatment procedure is completed, the chamber was cooled down under $N_2$ flow naturally before the sample was removed.

### Free-standing $TiO_2$ film preparation

To avoid the strong background signal from the Si wafer in STEM nano diffraction, free-standing $TiO_2$ films were prepared by ALD on a sacrificial PVP (polyvinylpyrrolidone) layer. 2% PVP aqueous solution was prepared and spin-coated on a Si wafer at 3000 rpm for 30 s. PVP-coated Si wafer was used in the ALD syntheses of amorphous TiO2 films under 100 °C for 200 cycles of deposition. Each cycle includes 0.5 s of $H_2O$ pulse + 60 s of $N_2$ purging + 0.5 s of $TiCl_4$ pulse + 60 s of $N_2$ purging. Accordingly, the additional 2400 cycles of water treatment were conducted after normal growth. After synthesis, samples were immersed in water at room temperature for 2 h to release the $TiO_2$ film. The free-standing $TiO_2$ films were scooped by TEM grids for STEM characterizations.

### PEC electrode preparation

Ni films were deposited on $TiO_2$-coated Si by sputtering using a CVC 601 DC sputtering system. The substrate was rotated at a speed of 5 rpm with argon flow at 25 sccm. The deposition was performed under 10 mTorr with 120 s deposition time. Then, the backside of the Si wafer were scratched by a diamond scribe and covered by Ga/In eutectic mixture. The silver paste was applied to fix the metal leads to the Ga/In eutectic mixture to achieve good Ohmic contact. After drying in a fume hood, the entire backside and partial front side of the Si/$TiO_2$/Ni electrodes were encapsulated by Epoxy (Loctite, 9460) with an exposed active area of ~0.05 $cm^2$. ImageJ were used to determine the exposed electrode area.

### Electrochemical characterizations

The PEC tests were carried out in a typical three-electrode electrochemical setup with Si/$TiO_2$/Ni as the working electrode, a Pt wire as a counter electrode, and a Hg/HgO electrode as a reference electrode. The electrolyte was 1 M KOH aqueous solution. For the cyclic voltammetry (CV) and chronoamperometry measurement, working electrodes were illuminated by a 150 W Xenon lamp coupled with an AM 1.5 global filter with a light intensity of 100 mW $cm^{-2}$ (one sun). Chronoamperometry curves were measured at a constant bias of 1.8 V vs. RHE. Electrochemical impedance spectroscopy (EIS) was conducted under open circuit voltage from 100 kHz to 0.1 Hz. All electrochemical curves were recorded using an Autolab PGSTAT302N station.

### Materials characterizations

Scanning electron microscopy (SEM) images were acquired on a Zeiss LEO 1530 field-emission microscope with a gun voltage of 5 kV and a working distance of ~3.5 mm. X-ray photoelectron spectroscopy was

acquired by Thermo Scientific K-alpha XPS instrument. Atomic force microscopy (AFM) topography was obtained using an XE-70 Park System. Device corrosion area percentage were statistically analyzed by ImageJ. Four-dimensional scanning transmission electron microscopy (4D-STEM) was performed using Thermo Fisher Scientific Themis Z STEM operated at 300 kV and equipped with an electron microscopy pixel array detector (EMPAD) to acquire nano-diffraction patterns from different sampling areas within the films[48]. The intensity variance of acquired nano-diffractions was calculated[49]. EDS was performed using FEI Themis Z microscope at 300 kV equipped with four Super-X detectors, and the chemical composition of amorphous films was obtained by analyzing EDS spectra using FEI Vlox software and Kα energies for Ti, Cl, and O. The presence of crystalline phases within the amorphous matrix was investigated by observation of film using low angle annular dark field (LAADF) STEM imaging, including diffraction contrast.

## Computational method

The amorphous $TiO_2$ structure was obtained through a melt-quenched process using the MA potential[50]. The structure was firstly melted in an NVT ensemble at 5000 K for 50 ps with a timestep of 0.5 fs, and cooled down to 3000 K with a timestep of 0.5 fs and 200 ps simulation time. Next, the model was equilibrated at 3000 K for another 50 ps. Finally, the model was annealed from 3000 K down to 300 K at a cooling step of 1 K/ps, equilibrated at 300 K for 100 ps, and statically optimized to minimal energy to obtain the final quenched atomic structure. The a-$TiO_2$ model consists of 87 atoms within a cubic box. The density of the structure model is consistent with the typical experimental density value of amorphous $TiO_2$ at room temperature (3.84 g/cm$^3$)[51]. The electronic property of amorphous $TiO_2$ with and without Cl ligands was then calculated using density functional theory (DFT) implemented in the Vienna ab initio simulation (VASP) package[52,53]. The generalized gradient approximation exchange-correlation functional Perdew, Burke, and Ernzerhof (PBE)[54] with the Hubbard U correction[55,56] was applied for the structure optimization and the density of states calculation. The $U$ value of 4.2 eV was applied to Ti atoms. The projector augmented wave method (PAW)[57] was used for the effective potential for all atoms. The PAW potentials used in these calculations have valence electron configurations of $3p^6 4s^2 3d^2$ for Ti, $2s^2 2p^4$ for O, and $3s^2 3p^5$ for Cl. The plane wave cutoff energy of 520 eV was used in all the calculations. The convergence criteria were $10^{-6}$ eV/cell for electronic self-consistent and 0.05 eV/Å for ionic relaxation, respectively. The Tetrahedron method with Blöchl corrections was applied for the density of states calculation.

## Data availability

All data from this study are available within the Article and its Supplementary Information. The raw data used in this study are available from the corresponding author upon reasonable request.

## Code availability

All processed data were available within the Article. The raw data used is available upon reasonable request.

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

## Acknowledgements

Research primarily supported by the US Department of Energy (DOE), Office of Science, Basic Energy Sciences (BES), under Award # DE-SC0020283.

## Author contributions

Y.D. and X.W. conceived the ideas and designed the experiments. X.W., D.M., and J.H. supervised the research. Y.D. and L.G. conducted ALD growth and device fabrication. Y.D. performed SEM, XPS, and electrochemical characterizations. M.A. carried out STEM and EDS mapping. J.M. conducted DFT calculations. C.C. performed AFM topography scanning. J.L. conducted Ni sputtering. Y.D. and Z.Z. performed XPS analysis. Y.D. and X.W. analyzed the data and wrote the manuscript. All authors discussed the experiment and contributed to the manuscript preparation.

## Competing interests

The authors declare no competing interest.
