## [Peer Review File · Nature Communications]

Substantial Lifetime Enhancement of Si-Based Photoanodes Enabled by Amorphous TiO₂ Coating with Improved StoichiometryREVIEWER COMMENTS

Reviewer #1 (Remarks to the Author):

The authors explored post-ALD in-situ water treatment can effectively decouple the ALD reaction completeness from crystallization. The treated TiO₂ film had a much lower residual Cl concentration, resulting in improved film stoichiometry and well-preserved uniform amorphous phase. As a result, the protected Si photoanode exhibited a substantially improved lifetime of up to 600 hours at a photocurrent density of 30 mA/cm². It can be potentially considered for publication after addressing the following comments.

1. The main argument in this work is that the decreased stability of TiO₂ stabilized Si photoanodes originates from the pin-hole formation in the TiO₂ film. But it is unclear why the pin-hole in TiO₂ would decrease the PEC performance as Si is an indirect band semiconductor and loss of the Si by corrosion would not deteriorate its ability of light absorption.
2. In Figure 5e, A few small pinholes could be presented on the surface of the reacted electrode for 250 h, but the stability was not affected until 600 h after that. It is known that when a pinhole occurs, corrosion is accelerated through the area without a protective layer (J. Phys. Chem. C 2012, 116, 19262–19267). How can the current not drop at all for a long period of time despite corrosion?
3. The existence of Cl in TiO₂ films can enhance the electrical conductivity of the films. With the post ALD treatment, how does the electrical conductivity of TiO₂ films change? Does it increase the on-set potential of PEC OER despite the better stability?
4. In Figure S8, the increase of time for water pulse caused a large number of nanoparticles on the surface. But water would react only with TiCl₄ precursor on the surface, and then unreacted water would be removed by purging. Please explain how the nanoparticles can be produced in detail.
5. Also, why didn't post-ALD water treated TiO₂ films have any nanoparticles unlike other TiO₂ films deposited using elongating water pulse time?
6. The LSV curves of the Si photoanode after > 30 h operation would be helpful to fully understand the PEC behavior related to figure 1e.
7. In Figure 2, appropriate characters (a, b, c) are not assigned to the descriptions of each figure.

Reviewer #2 (Remarks to the Author):

This paper demonstrates one step beyond the authors' previous efforts [citation 9] towards better stability of photoanodes by increasing photoanode lifetime from 500h to 600h, but the engineering goal of removing residual ligand during ALD growth while preventing the formation of polycrystalline during growth is very basic and the solution provided may also be specific to the TiCl₄ precursor, so I would not recommend publication.

Citation 41 of this paper provided a comprehensive analysis of the formation of polycrystalline during TiO₂ growth. In that paper, the authors proved that during thermal ALD growth with water, TiO₂ film

grown with TTIP shows reduced crystallinity, furthermore, they also explored how appropriate plasma can reduce crystallinity of TiO₂ films. For a more comprehensive study of improving the stability of TiO₂ coating, the authors of this paper could give a brief discussion of why TiCl₄ and water are the preferred precursors. Also, the authors could address if leftover ligands of other precursors would also affect the film stability in a similar fashion as Cl? Can leftover ligands of other precursors also be removed with additional water treatment as well?

As of current state of this paper, although I am convinced that the authors have improved the film quality for TiO₂ grown with TiCl₄ and water without plasma, I am not convinced that this is the best way to achieve the highest quality of TiO₂ film, and have some doubts if post-water treatment can help with other precursor based TiO₂ film growth as well.

Point-by-point responses to reviewers' comments:

Reviewer 1:

The authors explored post-ALD in-situ water treatment can effectively decouple the ALD reaction completeness from crystallization. The treated TiO₂ film had a much lower residual Cl concentration, resulting in improved film stoichiometry and well-preserved uniform amorphous phase. As a result, the protected Si photoanode exhibited a substantially improved lifetime of up to 600 hours at a photocurrent density of 30 mA/cm². It can be potentially considered for publication after addressing the following comments.

A: We thank the reviewer's high recognition of this work.

1. The main argument in this work is that the decreased stability of TiO₂ stabilized Si photoanodes originates from the pin-hole formation in the TiO₂ film. But it is unclear why the pin-hole in TiO₂ would decrease the PEC performance as Si is an indirect band semiconductor and loss of the Si by corrosion would not deteriorate its ability of light absorption.

A: We appreciate the reviewer's comment. In an as-fabricated Si/TiO₂/Ni photoanode, the Ni layer served as an electrocatalytic layer for water splitting. During PEC operation, pinholes formed in the TiO₂ layer and then evolved into large pores (shown in SEM images in **Figure 1f**). Therefore, the pores in TiO₂ would isolate the Si from the Ni catalyst layer. Meanwhile, OH⁻ ions penetrated through pinholes and oxidized Si into SiO_x. The insulating SiO_x layer would limit hole transport to the outer Ni catalyst. As a result, the Si/TiO₂/Ni photoanode PEC performance was impaired continuously. To clarify this relationship, we added "*The large pores would isolate the Si photoabsorber from Ni catalyst layer, and facilitate the formation of insulating SiO_x that limits the hole transport from Si to Ni catalyst. As a result, the Si/TiO₂/Ni photoanode PEC performance was impaired.*" (Page 5)

2. In Figure 5e, A few small pinholes could be presented on the surface of the reacted electrode for 250 h, but the stability was not affected until 600 h after that. It is known that when a pinhole occurs, corrosion is accelerated through the area without a protective layer (J. Phys. Chem. C 2012, 116, 19262–19267). How can the current not drop at all for a long period of time despite corrosion?

A: We thank the reviewer's comment. As we shown in **Figure 5e**, only a few small pinholes formed on the photoanode surface. Tiny pinholes could lead to the formation of insulating SiO_x around it and limit hole transport locally. This can be reflected by the anodic shift of the Ni(OH)₂/Ni(OOH) redox peaks and the decrease of J_{ph}-V curves slope. Because the water-treated TiO₂ film exhibited much higher chemical stability, the evolution from several pinholes to large interconnected pores was substantially suppressed. Therefore, Si photoabsorber and the TiO₂/Ni catalyst layer could still maintain a tight connection and allowed nearly impaired charge flow before the formation of large pores. To clarify the pinhole influences, we added the discussion: "*Owing to the much higher chemical stability of the water-treated TiO₂ films, the evolution from several pinholes to large interconnected pores was substantially suppressed. Therefore, Si*

photoabsorber and the TiO₂/Ni catalyst layer could still maintain a tight connection and allowed nearly impaired charge flow before the formation of large pores.” (Page 10)

3. The existence of Cl in TiO₂ films can enhance the electrical conductivity of the films. With the post ALD treatment, how does the electrical conductivity of TiO₂ films change? Does it increase the on-set potential of PEC OER despite the better stability?

A: We thank the reviewer’s comment. We studied the TiO₂ film conductivity using electrochemical impedance spectroscopy (EIS) shown in **Figure 5a** (shown as **Figure R1a** below). After partial Cl removal, water-treated TiO₂ kept a similar resistance compared to pristine TiO₂, suggesting partially removing Cl did not impair the charge transport property of the TiO₂ film. The similar conductivity between water-treated TiO₂ film and pristine TiO₂ film further determined the close onset potential and saturated photocurrent density in the J_{ph}-V curves in **Figures 5b**. The conductivity analysis was already discussed in the manuscript, Page 9: “Electrochemical impedance spectroscopy (EIS) of Si/TiO₂ electrodes was applied to study the possible TiO₂ film conductivity change induced by Cl removal (**Figure 5a**). The Nyquist plots were fitted based on the equivalent circuit model (Inset of **Figure 5a**). The Si/TiO₂ electrodes with and without water treatment showed similar charge transfer resistance at the high-frequency region, suggesting partially removing Cl did not impair the charge transport property of the TiO₂ film.” and “**Figure 5b** compares the J_{ph}-V curves of Si/TiO₂/Ni photoanodes with and without water treatment. Both curves exhibited similar onset potential and saturated J_{ph}.”

Figure R1 (Figure 5). **a**, EIS measurement of photoanodes protected by pristine and water-treated TiO₂ films under 1 sun illumination. Inset is the equivalent circuit for curve fitting. **b**, J_{ph}-V curves of Si/TiO₂/Ni photoanodes protected by pristine (black) and water-treated (red) amorphous TiO₂ films.

4. In Figure S8, the increase of time for water pulse caused a large number of nanoparticles on the surface. But water would react only with TiCl_4 precursor on the surface, and then unreacted water would be removed by purging. Please explain how the nanoparticles can be produced in detail.

A: We thank for the reviewer's comment. What described by the reviewer here is the ideal situation, where water molecules only reacted with chemisorbed TiCl_4 precursor by the ligand-exchange reaction. Practically, more kinetic processes may happen. Extended water pulse time may induce substantial surface diffusion of the as-deposited species that are loosely bonded to the surface. This may enable the rearrangement of surface atoms and form local nuclei on the surface.¹⁻⁴ Local nuclei can serve as TiO_2 growth seeds to evolve into nanoparticles as shown in **Figure S8**. Details explaining the formation of nanoparticles at extended water pulse time are added as "*However, the extended reaction time could also facilitate surface diffusion of as-deposited species that were loosely bonded to the surface. This may enable the rearrangement of surface atoms and form local nuclei on the surface,¹⁻⁴ which could serve as seeds for TiO_2 to grow into nanoparticles.*" (Page 11).

5. Also, why didn't post-ALD water treated TiO_2 films have any nanoparticles unlike other TiO_2 films deposited using elongating water pulse time?

A: We thank for the reviewer's comment. This is actually one of the key advantages of our post-ALD water treatment. As we explained in above response, extended water pulse time would facilitate surface diffusion, when the precursor molecules were not fully incorporated with the film. After ALD was completed, the 3D amorphous TiO_2 matrix was formed, and the completely coordinated Ti-O largely restricted their diffusion to form an ordered TiO_2 crystal lattice. Therefore, at the same temperature, post-ALD water treatment only extract the dangling or unreacted Cl ligand and remove them from the ALD film, but did not induce new crystallization of TiO_2 nanoparticles from its amorphous phase. This rationale is added in the manuscript, page 11: "*This was because when the ALD growth was completed, the 3D amorphous TiO_2 matrix was formed, and the completely coordinated Ti-O largely restricted their diffusion to form an ordered TiO_2 crystal lattice.*"

6. The LSV curves of the Si photoanode after > 30 h operation would be helpful to fully understand the PEC behavior related to figure 1e.

A: We thank for the reviewer's suggestion. The LSV curve of Si photoanode at 35 h is added as **Figure 1g** (also shown below in **Figure R2**). The onset potential shifted to 1.24 V versus RHE and the saturated photocurrent dropped to 21.9 mA/cm^2 , further validating the impaired hole transport and electrocatalytic performance due to corrosion of protective TiO_2 layer. We revise the descriptions: "*This onset potential was kept steady for the first ~15 hours, but quickly shifted positively to 1.11 V and 1.24 V versus RHE at the 30-hour and 35-hour operation time points, respectively (Figure 1g).*" (Page 5) "*Accordingly, saturated J_{ph} dropped continuously to 30.2, 29.8,*

28.8, and 21.9 mA/cm² at the PEC operation time of 5, 15, 30 h, and 35 h respectively, consistent with the decay trend in the stability test.” (Page 6)

Figure R2 (Figure 1g). J_{ph} -V curves of Si/TiO₂/Ni photoanode obtained at a series of time points from the chronoamperometry test.

7. In Figure 2, appropriate characters (a, b, c) are not assigned to the descriptions of each figure.

A: We thank for the reviewer’s suggestion. We carefully reviewed the manuscript and corrected the labeling of **Figure 2**.

Review 2:

This paper demonstrates one step beyond the authors’ previous efforts [citation 9] towards better stability of photoanodes by increasing photoanode lifetime from 500h to 600h, but the engineering goal of removing residual ligand during ALD growth while preventing the formation of polycrystalline during growth is very basic and the solution provided may also be specific to the TiCl₄ precursor, so I would not recommend publication.

A: We thank the reviewer for the comments, and carefully check of our previous publications. However, we respectively disagree that “the engineering goal of removing residual ligand during ALD growth while preventing the formation of polycrystalline during growth is very basic”.

Improving the lifetime from 500 hours to 600 hours and set a new record of PEC lifetime is not trivial. Fundamentally, we evidenced the significance of crystal-free amorphous phase and residue Cl ligands to the protection film stability in corrosive electrolyte. The understanding demonstrated here is transformative and can direct future research to further improve the lifetime and eventually reach the industrial requirement. We agree this residue removal strategy may not be general to all precursors. However, as we explained above, to create a general approach to removing residues from all possible ALD films or precursors is not the goal of this work. This strategy may be the best applicable only to small molecule Cl-based precursors, but the impacts to PEC, coating stability, and many other energy systems are still substantial, considering the broad application of amorphous TiO₂ as a surface coating.

Citation 41 of this paper provided a comprehensive analysis of the formation of polycrystalline during TiO₂ growth. In that paper, the authors proved that during thermal ALD growth with water, TiO₂ film grown with TTIP shows reduced crystallinity, furthermore, they also explored how appropriate plasma can reduce crystallinity of TiO₂ films. For a more comprehensive study of improving the stability of TiO₂ coating, the authors of this paper could give a brief discussion of why TiCl₄ and water are the preferred precursors.

A: We thank for the reviewer's comments. There are two main reasons for electing TiCl₄ precursor as the Ti source. First, the TiCl₄ precursor requires a relatively low growth temperature (<100 °C) to achieve sufficient Ti precursor dissociation during the ligand-exchange reaction. This low process temperature is critical to the growth of homogeneous amorphous TiO₂ film with minimal crystallization, as preferred by PEC protection. Other typical organometallic Ti precursors, including both titanium isopropoxide (TTIP) and tetrakis(dimethylamido)titanium (TDMAT) would suffer severe ligand residue contaminations at low temperatures.⁵⁻⁷ Therefore the recommended temperature commercial ALD of TiO₂ from those organometallic Ti precursors is typically above 150 °C. Second, the size of -Cl ligand is much smaller than the organic species such as isopropoxide ligand (TTIP) and dimethylamido ligand (TDMAT). The smaller ligand size can yield denser ALD TiO₂ film if a small amount of ligands presence.^{7,8} A denser film is preferred as a protection coating to limit ions transport, while a very small amount of residual ionic ligand is also helpful for charge transport. For oxygen sources, H₂O was a typical precursor for TiCl₄. Introducing high energy plasma can facilitate reaction completion, but also trigger more local crystallization instead of homogeneous amorphous TiO₂ film.^{9,10}

The reviewer pointed out the crystallization study in literature 41 (literature 45 in the revised manuscript).¹¹ We want to clarify that all the TiO₂ films presented in this reference are crystalline anatase, for both TTIP and TiCl₄ precursors. The authors demonstrated reducing plasma power could reduce the crystallinity, but it was clear that the crystalline phase was still significant in all the films. This work supports our statement that introducing plasma would facilitate crystallization. However, it did not show how to completely remove the crystalline phase and achieve amorphous TiO₂ films. This plasma influence has been discussed in the manuscript on page 11: *"introducing plasma, are always associated with undesirable crystallization, which also jeopardizes the protection lifetime as earlier research discovered."* In this revision, we added more discussions to explain why TiCl₄ and water were chosen as precursors for amorphous TiO₂ growth: *"Besides*

TiCl₄, tetrakis(dimethylamido)titanium (TDMAT) is another typical Ti precursor for ALD of TiO₂ films. However, growing TiO₂ by TDMAT at the same low temperature (100 °C) would leave a substantial amount of molecular ligand residues. These big residual molecules could reduce TiO₂ film density and even resulted in pores formation.^{46,47} (Page 12)

Also, the authors could address if leftover ligands of other precursors would also affect the film stability in a similar fashion as Cl? Can leftover ligands of other precursors also be removed with additional water treatment as well?

As of current state of this paper, although I am convinced that the authors have improved the film quality for TiO₂ grown with TiCl₄ and water without plasma, I am not convinced that this is the best way to achieve the highest quality of TiO₂ film, and have some doubts if post-water treatment can help with other precursor based TiO₂ film growth as well.

A: We thank the reviewer's comments and insights. As we reply to above comment, TiO₂ from TiCl₄ precursor is considered the best choice to achieve long protection lifetime. The reasons are their low deposition temperature and high ratio of amorphous phase (please see the details in our reply above). We want to highlight that this goal of work is to achieve amorphous TiO₂ films that can provide the longest PEC protection lifetime. Based on the reasoning stated above (e.g. other metal-organic precursors require higher ALD temperature, and already induced unwanted crystallization during the growth), it was less significant to examine whether this approach can apply to other precursors. This was the reason that we did not include other precursors in our original work.

However, from the reviewer's comments, we do reorganize the significant to compare different precursors in order to validate the significance of this work. In this revision, we grew TiO₂ films by ALD using TDMAT and H₂O as precursors, and investigated the water treatment and corresponding PEC performance. The ALD was conducted using a commercial Fiji G2 ALD system under the standard conditions (T=250 °C). The same water treatment at 100 °C was conducted, and the N concentration was measured to reflect the residual ligands change. The same PEC stability tests were also conducted to compare the lifetime to TiO₂ films grown by TiCl₄ and before and after water treatment. All the experimental details are added in the Method part in the revised manuscript. The results are illustrated below in **Figure R3**, as well as added to the supplementary material **Figure S13**.

SEM images in **Figure R3a** clearly shows that the TiO₂ film grown using TDMAT exhibited an appreciable amount of nanoparticles, suggesting crystallization already occurred during the growth. The 8-hour water treatment did not introduce obvious change to the surface nanoparticles (**Figure R3b**). From XPS spectra, a small N peak was detected from both TiO₂ films before and after water treatment (**Figure R3c**). Water-treated TiO₂ film showed a slightly decrease of N 1s peak from the unreacted TDMAT precursor, suggesting limited removal of residual N ligands. Similar to TiO₂ film obtained from TiCl₄, the fine Ti 2p spectra of both pristine and water-treated TiO₂ films made from TDMAT showed an almost identical shape with the dominating Ti⁴⁺ chemical state located at 459 eV (**Figure R3d**), implying the chemical state of Ti⁴⁺ cation in TDMAT-TiO₂ maintained after water treatment. By integrating the N, Ti peaks areas as a half-quantitative analysis, the N:Ti

ratio was found reduced by 13% (from 0.022 to 0.019), which was much lower than the reduction of Cl:Ti by 27 % (from 0.066 to 0.045). This limited residue removal could be attributed to the steric hindrance of the large dimethylamido ligand. Stability test showed that the TiO₂ film made from TDMAT had a lifetime of ~150 hours (**Figure R3e**), which was a little longer than the untreated TiO₂ film made from TiCl₄. However, the improvement of lifetime by water treatment was marginal, and only reached ~190 hours, much shorter than the ~600 hours offered by water-treated TiO₂ films made from TiCl₄. This result clearly evidenced that the TiO₂ films made from TDMAT may not be the optimal choice for surface protection. The pre-existence of crystalline nanoparticles brings inhomogeneity to the film and limits the lifetime improvement. Therefore, this result provides an additional support to our hypothesis that crystal-free is essential to achieve long protection lifetime, and small molecule TiCl₄ precursor is the optimal choice to achieve this goal.

To include this new result and discussion, a few sentences are added in revised manuscript: *“Therefore, ALD of TiO₂ was conducted under the recommended temperature (T=250 °C) using TDMAT as Ti precursor to understand the precursor relationship with the water treatment strategy and corresponding PEC performance. The as-grown TiO₂ film clearly showed an appreciable amount of nanoparticles on the surface, suggesting crystallization already occurred during the growth (**Figure S13a**). However, the water treatment did not introduce obvious change to the surface nanoparticles (**Figure S13b**). From XPS spectra, a small N peak was detected from both TiO₂ films before and after water treatment, with just an infinitesimal change of its intensity (**Figure S13c**). Integrating the N and Ti peak areas revealed a much smaller N:Ti ratio reduction of ~9% comparing to that of Cl:Ti ratio (27 %) from TiCl₄ precursor. This low removal ratio of TDMAT residues could be attributed to the limited diffusion as a result of the high steric hindrance from the large dimethylamido ligands. Accordingly, the lifetime of TiO₂ films made from TDMAT was only slightly extended from ~150 hours to ~190 hours after the water treatment (**Figure S13e**), much shorter than the ~600 hours offered by water-treated TiO₂ films made from TiCl₄. This result further evidenced that TDMAT may not be the optimal choice for ALD of protective TiO₂ films, as it introduces crystalline nanoparticles under conditions necessary to achieve sufficiently high ALD reactions. This result provided an additional support to our hypothesis that crystal-free is essential to achieve long protection lifetime, and small molecule TiCl₄ precursor is the optimal choice to achieve this goal.”* (Page 12-13)

Figure R3 (Figure S13). Si photoanode protected by pristine and water treated TiO₂ film from TDMAT precursor. a, b, Top view SEM images of pristine and water treated TiO₂ film on n-Si substrate. **c,** XPS N 1s peak of pristine and water treated TiO₂ film from TDMAT. **d,** XPS Ti 2p peak of pristine and water treated TiO₂ film from TDMAT. **e,** Chronoamperometry test of pristine and water treated TiO₂ protected Si photoanode measured in 1.0 M KOH aqueous solution under 1 sun illumination at an external bias of 1.8 V vs. RHE.

Reference:

- 1 Mackus, A. J., Verheijen, M. A., Leick, N. m., Bol, A. A. & Kessels, W. M. J. C. o. M. Influence of oxygen exposure on the nucleation of platinum atomic layer deposition: consequences for film growth, nanopatterning, and nanoparticle synthesis. *Chemistry of Materials* **25**, 1905-1911 (2013).

- 2 Mattinen, M. *et al.* Atomic layer deposition of crystalline molybdenum oxide thin films and phase control by post-deposition annealing. *Materials today chemistry* **9**, 17-27 (2018).
- 3 Aarik, J., Aidla, A., Mändar, H. & Sammelseg, V. J. J. o. C. G. Anomalous effect of temperature on atomic layer deposition of titanium dioxide. *Journal of Crystal Growth* **220**, 531-537 (2000).
- 4 Shi, J. *et al.* Electron microscopy observation of TiO₂ nanocrystal evolution in high-temperature atomic layer deposition. *Nano letters* **13**, 5727-5734 (2013).
- 5 Saari, J. *et al.* Tunable Ti³⁺-mediated charge carrier dynamics of atomic layer deposition-grown amorphous TiO₂. *The Journal of Physical Chemistry C* **126**, 4542-4554 (2022).
- 6 Reiners, M. *et al.* Growth and crystallization of TiO₂ thin films by atomic layer deposition using a novel amido guanidinate titanium source and tetrakis-dimethylamido-titanium. *Chemistry of Materials* **25**, 2934-2943 (2013).
- 7 Dufond, M. E. *et al.* Quantifying the extent of ligand incorporation and the effect on properties of TiO₂ thin films grown by atomic layer deposition using an alkoxide or an alkylamide. *Chemistry of Materials* **32**, 1393-1407 (2020).
- 8 Sinha, A. *et al.* Area selective atomic layer deposition of titanium dioxide: Effect of precursor chemistry. *Journal of Vacuum Science & Technology B: Microelectronics and Nanometer Structures Processing, Measurement, and Phenomena* **24**, 2523-2532 (2006).
- 9 Saari, J. *et al.* Low-Temperature Route to Direct Amorphous to Rutile Crystallization of TiO₂ Thin Films Grown by Atomic Layer Deposition. *The Journal of Physical Chemistry C* **126**, 15357-15366 (2022).
- 10 Chaker, A. *et al.* Understanding the mechanisms of interfacial reactions during TiO₂ layer growth on RuO₂ by atomic layer deposition with O₂ plasma or H₂O as oxygen source. *Journal of Applied Physics* **120**, 085315 (2016).
- 11 Chiappim, W. *et al.* Relationships among growth mechanism, structure and morphology of PEALD TiO₂ films: the influence of O₂ plasma power, precursor chemistry and plasma exposure mode. *Nanotechnology* **27**, 305701 (2016).

REVIEWERS' COMMENTS

Reviewer #1 (Remarks to the Author):

I'm satisfied with the revision and therefore recommend publication in Nature Communications.

Reviewer #2 (Remarks to the Author):

After the revision I am more convinced that the authors are demonstrating the state of the art deposition method of amorphous TiO₂ film in this paper, and considering the broad application of amorphous TiO₂ coatings, this paper can be considered for publication. I apologize for not carefully surveying the lifetime of TiO₂ films deposited with other precursors in my original comments, but the added comparison of TiO₂ film lifetimes with different precursors have better convinced me, and hopefully other readers who are used to precursors other than TiCl₄, that the deposition method shown in this paper is the better way for achieving amorphous TiO₂ coating with high stoichiometry.